# Which Suture to Choose in Hepato-Pancreatic-Biliary Surgery? Assessment of the Influence of Pancreatic Juice and Bile on the Resistance of Suturing Materials—In Vitro Research

**DOI:** 10.3390/biomedicines10051053

**Published:** 2022-05-02

**Authors:** Marcin Gierek, Katarzyna Merkel, Gabriela Ochała-Gierek, Paweł Niemiec, Karol Szyluk, Katarzyna Kuśnierz

**Affiliations:** 1Center for Burns Treatment im. Dr Sakiel, ul. Jana Pawła II 2, 41-100 Siemianowice Śląskie, Poland; 2Institute of Materials Engineering, Faculty of Science and Technology, University of Silesia, ul. 75. Pułku Piechoty, 41-500 Chorzów, Poland; 3Dermatology Department, City Hospital in Sosnowiec, ul. Zegadłowicza 3, 41-200 Sosnowiec, Poland; g.ochala@wp.pl; 4Department of Biochemistry and Medical Genetics, Faculty of Health Sciences in Katowice, Medical University of Silesia in Katowice, 40-752 Katowice, Poland; pniemiec@sum.edu.pl; 5Department of Physiotherapy, Faculty of Health Sciences in Katowice, Medical University of Silesia in Katowice, 40-752 Katowice, Poland; karol.szyluk@sum.edu.pl; 6Department of Orthopaedic and Trauma Surgery, District Hospital of Orthopaedics and Trauma Surgery, 41-940 Piekary Śląskie, Poland; 7Department of Gastrointestinal Surgery, Medical University of Silesia in Katowice, ul. Medyków 14, 40-752 Katowice, Poland; kasiachir@wp.pl

**Keywords:** absorbable sutures, bile, pancreatic juice

## Abstract

(1) Background: The choice of appropriate surgical suture during operation is of great significance. Currently, there are no objective studies regarding the resistance of commonly used sutures in biliary tract surgery. (2) Methods: This fact leads one to conduct research concerning the resistance of the sutures (Polydioxanone, Poliglecaprone, Poliglactin 910, and their analogues coated with antibacterial triclosan) in the environment of sterile and contaminated bile and pancreatic juice. Tensile strength was tested at days 0, 7, 14, 21, and 28 of research. The study was performed in in vitro conditions for 28 days. (3) Results: Pancreatic juice and bile has a significant influence on the tensile strength of each suture. (4) Conclusions: The study indicated that sutures made of polydioxanone had the best qualities during the entire experiment.

## 1. Introduction

A procedure with a particularly high complication and mortality risk is pancreatoduodenectomy (PD). This procedure is mainly performed due to tumors within the pancreato–duodenal area. There are many factors influencing the occurrence of a puncture in the pancreato–duodenal anastomosis, such as the emergence of a pancreatic fistula after PD (POPF). It occurs in 5–30% cases [1,2]. There are also other important factors influencing the complication, such as the diameter of the main pancreatic duct, and pancreatic texture [3,4].

One may also encounter comorbidities such as malnutrition, inflammation, radiation, chemotherapy, diabetes, infections of the bile following bile duct stenting, intraoperative blood loss, the reconstructive method (pancreatojejunostomy, pancreatogastrostomy), and the anastomosis technique (duct-to-mucosa, invagination pancreatojejunostomy techniques); however, dehiscence of the biloenteric anastomosis and consequent biliary fistula are quite rare (1–9%) [5].

When operating on the pancreas, the choice of a suitable suture is one of the conditions that is vital for the success of the procedure. Each type of suture has its own mechanical features and produces different tissue responses [1]. The ideal material should be non-traumatic, incite minimal inflammation, and preferably be absorbed over a suitable time frame which avoids anastomotic breakdown in situations where healing may be delayed. Intestinal anastomoses (biliary–intestinal, intestino–intestinal, pancreato–intestinal) are exposed to the action of bile and pancreatic juice. The pair influences the sutures, and therefore, accelerates the degradation of surgical material. In the literature, one may encounter data concerning the premature dehiscence of the sutures in intestinal anastomoses exposed to pancreatic juice and bile [6,7,8,9,10,11,12].

Infection of the bile and pancreatic juice frequently results in difficulties in the healing process. There is not much clinical and experimental research concerning the suture used in pancreatic and bile duct anastomoses in the intestine [1,2,4,13,14,15]. The difficulty of the clinical research represents an inability to estimate if the suture is degraded or intact at the point of the leak. An assessment may be performed only during re-laparotomy, if the area is accessible at examination.

The aim of the present research is to determine the resistance of absorbable sutures in the environment of sterile and contaminated bile, and sterile and contaminated pancreatic juice in relation to the exposure time to the fluids. Moreover, the aim is to determine the influence of the antibacterial coating on the resistance of the examined sutures.

## 2. Materials and Methods

Three types of absorbable sutures (PDS, Vicryl, Monocryl), as well as their analogues with an antibacterial triclosan coating (PDS plus, Vicryl Plus, Monocryl Plus), were used in the experiment. Table 1 shows the chemical structure and the general characteristics of absorbable surgical sutures with and without an antibacterial coating. The materials were immersed in five environments (sterile pancreatic juice, contaminated pancreatic juice, sterile bile, contaminated bile, and physiological saline).

Bile and pancreatic juice were obtained from one patient undergoing pancreatic tumor treatment. Pancreatic juice was collected fourfold (during four postoperative days) from a drain inserted into the Wirsung’s duct. Bile, similarly, was collected fourfold (four postoperative days) from the Kehr’s drain inserted into the common bile duct. Biological material was collected into sterile cryotubes (Corning^®^ Internal Threaded Polypropylene Cryogenic Vial, Corning, Somerville, MA, USA) of 5 mL volume. Both the bile and pancreatic juice were distributed in sterile conditions.

Sterility of the material was bacteriologically confirmed prior to freezing (microscopic examination and microbiological culture collected after four days from each sample). A microscopic examination confirming the presence of the bacteria excluded the material from the freezing process. The culture obtained after four days confirmed the sterility of the frozen material. If the culture was positive, the material was excluded as well.

Amylase and lipase levels in pancreatic juice before freezing and after thawing, along with pH measurements of pancreatic juice and bile, are shown in the Appendix A.

Moreover, prior to freezing, with the use of the laboratory pH meter (Piccolo HI 98111, Hanna Instruments S.R.L., Woonsocket, RI, USA), the pH of bile was measured. The biological material in cryoprobes was frozen and stored in a laboratory freezer (Revco™ High-Performance Lab Freezers ULT430A, Thermo Fisher Scientific, Waltham, MA, USA) at a temperature of −20 °C [16]. Two methods of bacteriological examination were performed, including a microscopic examination of the material (Kern OBN-14, Kern Optics, Balingen Germany), and if a high-power field presence of bacteria in bile was noted, the probe was excluded from the test. Additionally, a microbiological culture was performed. Growth media for the bacteria, fungi, and mold were used. The media used in the study were: MacConkey, Kliegler, Clauber, Chapman, Sabouraud (Merck KGaA, Darmstadt, Germany).

If the results of both the microbiological and microscopic examination were negative, the material was considered sterile. A similar means of bacteriological examination was introduced after defrosting the material. Attenuated material was obtained by contamination of sterile pancreatic juice and bile with (1) *Escherichia coli*, (2) *Klebsiella* spp. and (3) *Enterococcus faecalis* (bacteria most frequently occurring in pancreatic and biliary tract infections) [17,18,19,20,21].

For the purpose of contamination, three bacterial suspensions were prepared (consisting of sterile saline and particular bacterial strain). These were examined according to the MacFarland 0.5 standard with turbidity meter (MicroScan TurbidityMeter, Siemens AG, Munich, Germany) [22]. Following the turbidimetric examination, using a pipette (Eppendorf Xplorer^®^, Eppendorf AG, Hamburg, Germany), an amount of 10 µL of each suspension was collected and added to probes containing sterile bile and sterile pancreatic juice. The sutures were immersed in the biological material and incubated in a laboratory incubator at a temperature of +37 °C.

Twenty-four pieces of PDS, Monocryl, Vicryl, PDS Plus, Monocryl Plus, and Vicryl Plus were inserted to the six tubes of defrosted bile. The probes with the sutures were incubated in a laboratory incubator (Hanna^®^ COD Test Tube Heater, HI839800-01, Hanna Instruments S.R.L., Woonsocket, RI, USA) at a temperature of +37 °C. Analogous conduct was introduced in the environment of pancreatic juice.

Resistance measurements of the sutures were performed on a tensile testing machine INSTRON 4469 (Instron^®^, Norwood, MA, USA). The examination of the material resistance to the environment was performed on days 0, 7, 14, 21, and 28 of the immersion of the sutures. The tensile strength (Rm) was calculated as a ratio of the measured tensile force obtained during the static tensile test, in relation to the original cross-sectional area of the sutures. Day “0” is the date of the measurement of a suture collected directly from a package (baseline, initial state). At specific dates, six pieces of the sutures were removed from the test tube and the pieces’ resistances were examined. The bile and pancreatic juice were exchanged daily. Figure 1 shows the stages of the experiment.

A database of the clinical material was created under a licensed version of an Excel spreadsheet v. 2003 (Microsoft, Redmond, WA, USA). The data was implemented into a Statictica package v. 7.1 (Statsoft, Tulsa, OK, USA) and statistical software MEDCALC v. 11.3.1 (MedCalc Software Ltd., Ostend, Belgium). At the first stage, one has to estimate the basic characteristics of the descriptive statistics of the resistance of examined sutures including: the arithmetic mean of the tensile strength, the median, maximum, and minimal value, quartile 25% (lower), quartile 75% (upper), standard deviation, standard error of the mean SEM, kurtosis, and skewness. A detailed summary of the statistical evaluation for both the reference surgical sutures (Appendix A) and the tested sutures (Appendix A) during the experiment are presented in the Appendix A.

In the statistical evaluation, one accepted the significance level of *p* < 0.05. Many parametric tests require that the data come from a near-normal distribution, therefore, it was important to perform a test examining the normality of the distributions. Due to the small number of trials, which results from the availability of the sutures on the market (all possible ones were considered), we performed the Shapiro–Wilk Test (S–W). The S–W test is the preferred test to examine the normality of a probability distribution because of its strong potency compared with other available tests.

In the next stage of the statistical evaluation, the following tests were used: 1. the Levene test, to verify the hypothesis of the homogeneity of variance; 2. the ANOVA one-way test to verify the hypothesis of the equality of means among the groups; 3. the post hoc test, which demonstrated a reasonably significant difference; 4. the Student *t*-test for the two means; 5. the Fisher homogeneity variance test which showed that the variance is not homogeneous, along with the Saterthwaite test.

A scanning electron microscope (SEM) was used for observations of the surface of investigated sutures immersed for 28 days in the sterile and contaminated pancreatic juice, bile, and saline solutions. We used a HITACHI S-3400N (Hitachi Ltd., Tokio, Japan) microscope with the possibility of magnification 5× to 100,000×. This device made it possible to observe the surface of non-conductive materials (low vacuum) equipped with X-ray spectrometers: Thermo Noran EDS and Thermo MagnaRay WDS (EDS-Energy-dispersive, WDS Wavelength-dispersive spectroscopy), with a backscattered electron diffraction detector (EBSD). The sutures’ images were obtained from backscattered electrons (BSE). All observations and image acquisitions were made at 15 kV accelerating voltage.

## 3. Results

In the saline solutions, the tensile strength (Rm) of each suture until day 28 was able to be determined. Figure 2 shows the comparison of the tensile strength of each surgical suture examined in sterile saline at day 7 and 28 of the experiment. It was noted that in the biological environments, only PDS and PDS Plus sutures were able to endure, and one was able to determine the tensile strength until the termination of the experiment (day 28). Other suture materials underwent destruction after a 21 day exposure period to pancreatic juice and bile. The final resistance measurement for each of the sutures in the biological material was obtained on day 21 of the experiment.

### 3.1. Environment of Bile and Pancreatic Juice

There are statistically significant differences on the 7th, 14th, and 21st day of the exposition. Figure 3 shows the comparison of the tensile strength of surgical sutures examined in sterile and contaminated bile on day 7. We observed a decrease in tensile strength for all polymers in the contaminated environment compared with the sterile one. The greatest changes were observed for the Vicryl and Vicryl Plus polymers. On day 7 of the experiment, it was observed that the sutures, Vicryl and Vicryl Plus, are characterized by the highest endurance in both (sterile and contaminated) environments. We observed a much higher value of tensile strength in the sterile environment than in the contaminated one (*p* < 0.05 see Appendix A). At that point of the examination, one proved to be a crucial influence on the contaminated environment with regard to the decrease in material resistance of Vicryl sutures. Additionally, one notes that the contaminated environment has more influence on the decrease of the resistance of each of the sutures.

Figure 4 demonstrates enormous differences in material resistance to the environment at day 21 of the examination. One has demonstrated that the exposure time evidently influences the decrease in the tensile strength of the sutures (Vicryl and Vicryl Plus on day 21 of the examination, which showed a twofold decrease in comparison with day 7 of the examination). Moreover, a crucial influence on the contaminated environment, with regard to the degradation of surgical sutures on day 21 of the examination, was indicated (Figure 4). A significantly increased degradation (environmental resistance decrease) of the sutures immersed in pancreatic juice was noticed. Pancreatic juice particularly influenced the resistance of Vicryl and Vicryl Plus sutures, as well as Monocryl and Monocryl Plus on day 21 of the experiment (Figure 4), wherein a noticeable, greater decrease in the tensile strength of PDS sutures in bile compared with pancreatic juice environment was shown.

When analyzing the tensile strength values of Monocryl sutures, one noted statistically significant differences on day 21 of the exposure, wherein the Rm Level of the Monocryl suture in the environment of sterile bile is higher than in the environment of contaminated bile (*p* = 0.0153, see Appendix A). Thus, it is a confirmation that surgical sutures undergo more destruction in the contaminated environment than in the sterile one.

On day 21 of the exposure (Figure 5), the level of PDS suture resistance in the sterile environment of bile is only slightly higher than in the contaminated one (*p* = 0.0172, see Appendix A). In the case of pancreatic juice, much higher tensile strength values were observed for an infected environment than for a sterile one. When analyzing the tensile strength values, one may find that the decreased resistance of PDS in the environment is statistically insignificant in comparison with the initial values. Resistance of PDS is basically constant.

### 3.2. Scanning Electrone Microscope

SEM observations of the surface images of the tested surgical sutures immersed in a sterile and contaminated environment also confirm that the surface of the sutures in infected mediums are more degraded than those in the saline medium. Figure 6 shows scanned electron microscope images of uncoated sutures immersed during days 21 and 28 of the experiment in various environments. In the case of surgical sutures made of Vicryl and Monocryl polymers, it was not possible to perform surface structure studies after 21 days of the experiment due to material degradation. Observations under electron microscopy regarding the suture surface did not reveal a disintegration in the architecture of the PDS and Vicryl suture materials. Clear degradation of the structure of the sutures can be observed for monofilament sutures made of Monocryl in comparison with the others. Monocryl sutures exposed to pancreatic juice demonstrated many more fractures and greater surface disintegration than in case of bile. In the case of the sutures made of Vicryl, after 21 days of the experiment, a distinct unraveling in the multifilament sutures was observed; Karaman et al. obtained similar results [15].

## 4. Discussion

The physiological healing phases of the gastrointestinal anastomosis, in which several growth factors and extracellular matrix proteins are active, are of great importance [23]. Parts of the anastomotic healing process consist of the exudative–inflammatory (0–4 days), proliferative (4–14 days), and repair–remodeling (14–180 days) phases [24]. The exudative–inflammatory phase is the weakest phase in which the anastomosis shows a greater propensity for leakage, culminating in a complete breakdown under tension. In the proliferative phase, the wound is finally closed, and different types of cells lead to the reproduction of the extracellular matrix, angiogenesis, and re-epithelialization. At this point, it is worth considering how the strength of the investigated materials changed in the first two weeks of the experiment. Initially, the highest tensile strength was demonstrated by a Poliglactin 910 multifilament surgical suture (Vicryl) (293 MPa) and a monofilament suture made of Poliglecaprone 25 (Monocryl) (236 MPa); however, the initial tensile strength of poly-p-dioxanone (PDS) was much lower (around 117 MPa). After 7 days of the experiment, monocryl sutures showed a decrease in tensile strength by approximately 50% in both the sterile and the contaminated environments; however, after 14 days, 80% of the monocryl was degraded in the infected environment with the pancreatic juice. Vicryl surgical sutures were degraded by 40% after 7 days, and by 80% after the 14th experiment in the contaminated pancreatic juice. Surgical sutures made of PDS polymer that was exposed to pancreatic juice in the sterile environment decreased their tensile strength by 15%; however, in the infected environment, no more than 3% of the strength degraded after two weeks. It can be said that the hydrolysis of the PDS polymer was slower in the contaminated environment. Although PDS material has the lowest initial tensile strength compared with the other sutures, it did not lose its mechanical properties throughout the entire experiment.

Bile and pancreatic juice have a significant effect on the strength of surgical sutures tested in vitro. The highest statistically significant decrease in strength was shown in the sutures of Poliglecaprone 25 and Poliglactin 910 in the contaminated pancreatic juice.

In conclusion, after 21 days of the experiment, for surgical sutures made of the polymers Monocryl, Monocryl Plus, Vicryl, and Vicryl Plus, it was not possible to perform the mechanical strength tests due to very high degradation of the sutures. Only in the case of the PDS and PDS Plus polymers, could the tensile strength test be carried out at the final stage of the experiment (i.e., after 28 days). Table 2 summarizes the percentage changes in the strength values with respect to the Rm values for the surgical reference sutures over the course of the entire experiment for different environments.

On day 21, it is noticeable that the greatest decrease in tensile strength is for Monocryl and Vicryl and their triclosan analogues in infected pancreatic juice. It is noticeable that a decrease in tensile strength for Monocryl on day 21 is 94.46% in infected bile and 97.90% of their baseline strength. The decrease for Vicryl on day 21 is 85.39% in infected bile and 94.74% in infected pancreatic juice. The decrease in tensile strength for PDS on day 21 is 8.96% in infected bile and 26.96% of their baseline strength in infected pancreatic juice (Table 2).

We could observe that the PDS sutures manifested the maximum resistance in the infected environment, both in the case of bile and pancreatic juice.

Muftuoglu et al. also noticed that PDS sutures were characterized by a minimal decrease in the resistance measurement values [25]. Tian et al. suggests that during pancreatic and biliary duct surgery, one should use a suture which decomposes slowly, without sudden changes in resistance [12].

One cannot disagree with the statement; surgical sutures should be characterized by an extended time of the tissue suspension. Our study proved that PDS and PDS Plus conform to the standard.

Freudenberg et al. noticed that significant decreases in the resistance of Vicryl sutures occur in pancreatic juice and bile, but PDS is very stable in both environments [16]. Our examination confirms these results. PDS and PDS Plus were the only sutures with determinable resistance on day 28 of the experiment. Other sutures were completely destroyed and neutralised after day 21 of the examination.

Chung et al. noticed that the presence of *E. coli* bacteria in vitro in bodily fluids influences the resistance of the sutures and results in the decrease of the durability [26]. Our study also definitely substantiates these results. It should be noted that there was a statistically important decrease in resistance in contaminated environments in comparison with the sterile environments.

In our study we compare monofilament sutures (PDS, PDS Plus, Monocryl, Monocryl Plus) and multifilament sutures (Vicryl, Vicryl Plus). There are many clinical differences in the use of monofilament and non-monofilament sutures. Non-absorbable multifilament sutures have a higher incidence of wound infection and sinus formation [27].

The absorbable sutures with a slightly longer half-life (Vicryl) are less affected by exposure to bile. As the absorbable sutures are braided to overcome the rigidity that exists in their monofilament form, they have the potential of inciting a more intense inflammatory response than would be expected from a monofilament suture [28]. The emergence of monofilament absorbable sutures with a longer half-life, such as PDS, offers surgeons the potential of an ideal suture material for bile duct anastomoses [28].

Monofilament sutures (polydioxanone) have less variation after exposure to pancreatic juice and bile [13,25]. Our study confirms that PDS, which is a monofilament, is the most stable material to reduce strength. Vicryl sutures (non-monofilament) decomposed on day 21 of the study; however, taking into account the differences in the structure of polymers, monofilament threads seem to be better in terms of anastomoses exposed to bile and pancreatic juice.

Braided sutures are generally less elastic and have less memory than monofilament sutures. Sutures with less memory often provide greater knot tightness and security [1]. Early results show that polydioxanone is the only type of suture that is able to keep its original tensile strength after incubation in pancreatic juice and bile. Among non-absorbable sutures, silk maintains a good level of tensile strength, although it is worse than polydioxanone. In the Andrianello series, the possible suture that induced damage on the pancreatic remnant (expressed by post-operative hyperamylasemia) was similar to monofilament polydioxanone and braided polyester (23.8% vs. 21.4%; *p* = 0.7) [1].

In contrast to polypropylene—due to its long-lasting absorbable capacity—polydioxanone is not retained permanently as a foreign body [14]. Braided multi-filament sutures (polyglactin 910 and silk) may be advantageous over monofilament sutures in terms of knotting safety and approximation of wound edges under a certain tension without causing capsular or parenchymal tears, particularly in soft pancreatic texture [29]. In addition, polyglactin 910 causes less inflammation in surrounding tissues than silk [14].

Leaks in pancreato–intestinal anastomosis most commonly occur in the early postoperative stage, within 10 days [2]. The healing process consists of three stages, of which the first is an inflammatory phase that lasts between 4–7 days. The role of the suture material used is of utmost importance at the time and relies on closing and sustaining the anastomosed tissues. If the healing process is faultless, in the following (proliferative) phase, the role of the sutures is minor. Clinical experiments suggest that the early healing phase is crucial, although leakages may occur in the following stage which may result in complications [30].

The occurrence of pancreatic or bile fistula results in major complications (peritonitis, intraabdominal abscess, pseudoaneurysm, hemorrhage) and can greatly prolong the healing process [4]. Taking this into consideration, the healing time of the anastomoses is greatly extended in comparison with physiological healing; therefore, it is reasonable to suggest the use of sutures which can sustain a longer-lasting period. Leaks concerning only a part of the anastomosis periphery, which are mainly dealt with using conservative treatments or non-invasive methods, are more advantageous compared with complete dehiscence anastomosis, which can result in surgical intervention. One may conclude that if complications occur while manipulating within an anastomosis area, the remaining sutures may decrease the consequences of mechanical damages. One of the means of decreasing the number of leakages of the pancreato–intestinal anastomosis is an internal or external stenting of the anastomosis. In this case, installing a stent in the Wirsung duct requires using sutures to fasten the stent to the pancreatic and/or intestinal parenchyma. Due to the fact that there is no healing process, only mechanical clamping and the time when the stent was installed are determined by the length of the degradation time of the suture. If there is a leak, the aim is to prolong the stent durability. As the pancreas is an extremely fragile organ which is susceptible to trauma that can result in acute pancreatitis, in order to decrease the trauma resulting from surgery, one has to limit the number of the sutures. The use of thin sutures and seams with little tension is therefore recommended [31]. There is no consensus concerning whether the use of simple, interrupted sutures or continuous sutures is more advantageous. According to some authors, continuous sutures in pancreatic anastomosis effectively reduces the POPF [32]. Stenosis of pancreatico–enteric anastomosis is most often a consequence of postoperative complications such as pancreatic fibrosis, a thin Wirsung duct connected with the intestine, and acute postoperative inflammation of the left pancreatic stump [32]. Stenosis of the pancreatico–enteric anastomosis following pancreatoduodenectomy (PD) is a late post-operative complication appearing after a median of 18 (8–120) months [33]. Some of the reports advocate for the advantages of continuous sutures in anastomoses including less time and less damage to the pancreas and a lower rate of pancreatic fistula formation [34,35,36]. Continuous sutures have a lower incidence of suture line stenosis because of the expansion and contraction of the intraluminal force, and less muscular fibrosis from tissue ischemia reacting with suture material. Most of the techniques for pancreatic anastomosis are based on the use of single interrupted sutures [37].

The majority of the pancreato–intestinal anastomoses use single interrupted sutures. The abovementioned technique is guaranteed to sew and close the wound under determined tension without tearing the parenchyma, particularly in the areas with a soft pancreatic texture. This study brings us closer to an answer to the question of which sutures should be chosen in the anastomosis exposed to bile and pancreatic juice. If one suspects that healing will be demanding and prolonged (and may result from i.e., the patient’s different comorbidities) it is advisable to elect a suture which has the ability to sustain prolonged tension (in in vitro conditions, it ought to be the suture with the last determined resistance). The results of the present research indicate that the sutures made of polydioxanone are characterized by proprieties that were already mentioned. One has not indicated the advantage of sutures coated with an antibacterial layer.

The study proves that the choice of the surgical material is of greater importance (polymer the suture is made of) than the presence of an antibacterial layer in terms of sustaining the resistance of the material in sterile and contaminated environments. It seems that the results of the current study can help in selecting the right suture for the anastomosis performed, which will result in fewer complications resulting from an unfortunate choice of material.

## 5. Conclusions

The presence of contaminated environment of pancreatic juice and bile has a significant influence on degradation of all the sutures. Resistance of the sutures depends on the exposure time and environment. For the sutures, the environment of contaminated pancreatic juice is the most aggressive. An antibacterial coating does not influence the resistance of the suture. In the present study, Polydioxanone (PDS) is the material which has the best resistance and the longest degradation time.

## Figures and Tables

**Figure 1 biomedicines-10-01053-f001:**
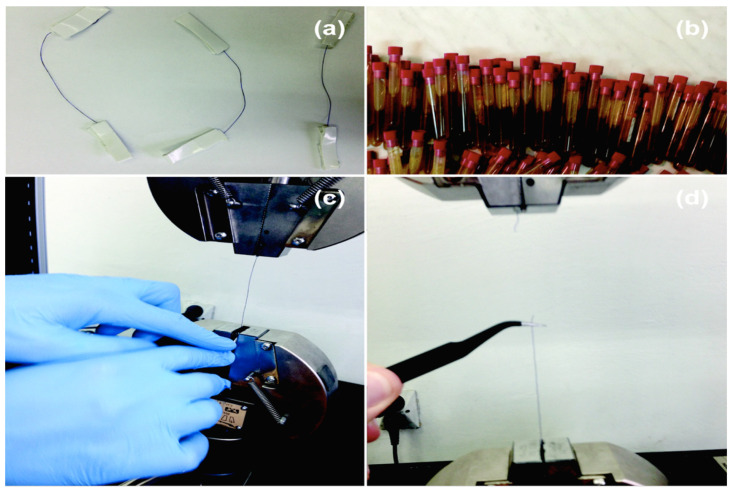
Stages of the experiment: (**a**) sutures before the stretch testing; (**b**) collected sterile bile and sterile pancreatic juice in cryotubes; (**c**) INSTRON tensile machine; (**d**) suture after stretch test.

**Figure 2 biomedicines-10-01053-f002:**
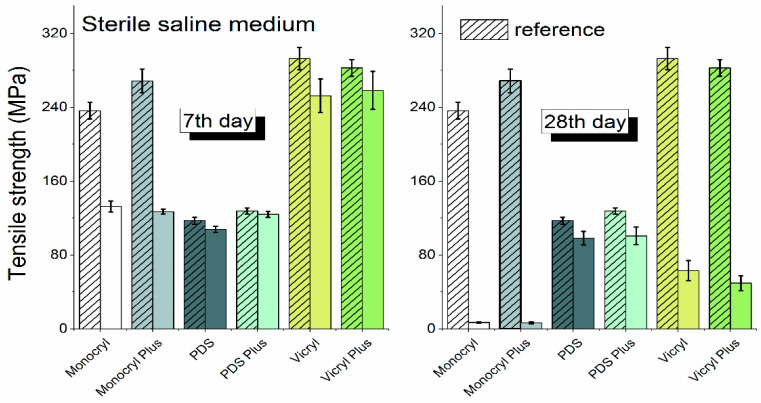
Comparison of the tensile strength of each surgical suture examined in sterile saline on days 7 and 28 of the experiment.

**Figure 3 biomedicines-10-01053-f003:**
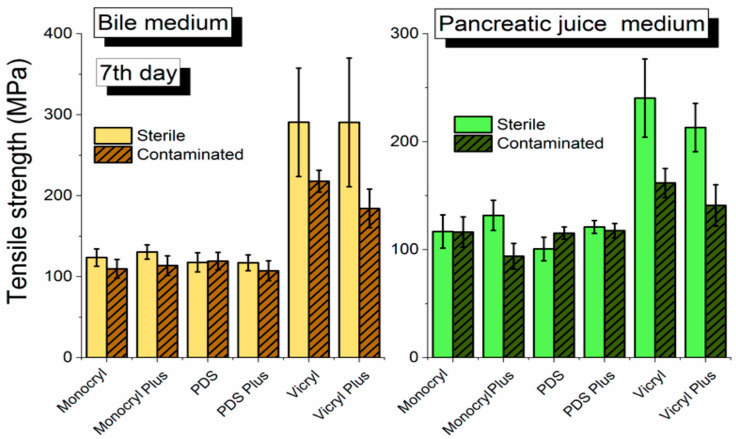
Comparison of the tensile strength of surgical sutures examined in sterile and contaminated bile at day 7 of the experiment.

**Figure 4 biomedicines-10-01053-f004:**
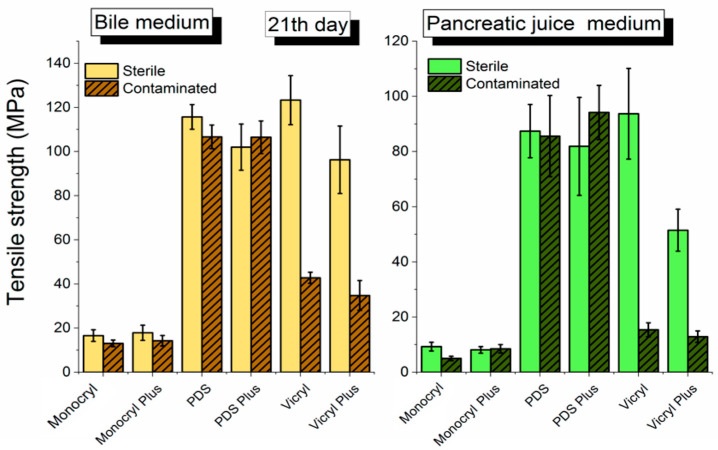
Comparison of the tensile strength of surgical sutures in bile and pancreatic juice (sterile and contaminated material) on day 21 of the experiment.

**Figure 5 biomedicines-10-01053-f005:**
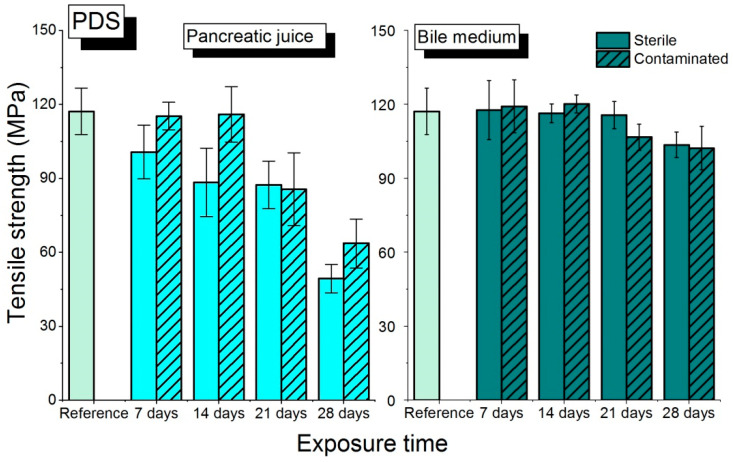
The results of the tensile strength of PDS sutures in bile and pancreatic juice, sterile and contaminated, in relation to the exposure time.

**Figure 6 biomedicines-10-01053-f006:**
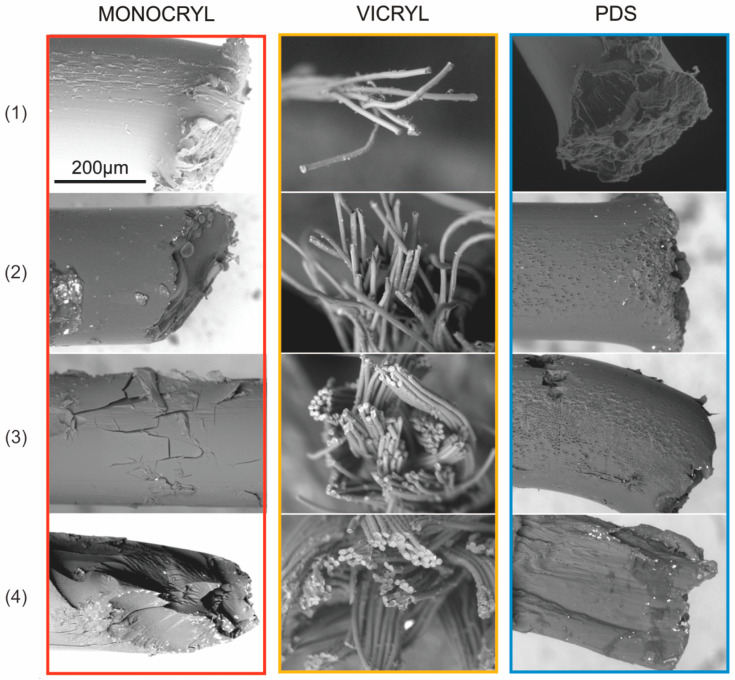
Scanning electron microscope images of uncoated sutures on day 21 (for Monocryl and Vicryl in sterile and infected pancreatic juice and bile) and day 28 of the experiment (for all sutures in a saline medium, PDS in all mediums): red box—Monocryl suture; orange box—Vicryl suture; blue box—PDS suture: (**1**) reference SEM images for the starting sutures (no medium); (**2**) saline—sterile environment; (**3**) pancreatic juice—contaminated environment; (**4**) bile—contaminated environment.

**Table 1 biomedicines-10-01053-t001:** Characteristics of absorbable surgical sutures with and without an antibacterial coating (data from suture manufacturer’s leaflet: Ethicon, Johnson&Johnson).

Name and Chemical Structure of Materials	Suture Trade Names	Tissue Support Profile [%]	Period of Maintaining the Tissue Tension of Sutures	Absorption Period
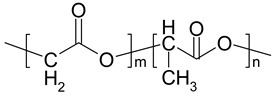 Poly(glycolide/ε-caprolactone) Copolymer or Polyglecaprone 25	Monocryl	60%—7 days 30%—14 days	21–28 days	90–120 days
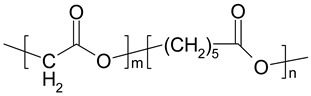 Poly(glycolide/L-lectide) Copolymer or Polyglactin 910 (m = 90, n = 10)	Vicryl	75%—14 days 50%—21 days	28–35 days	56–70 days
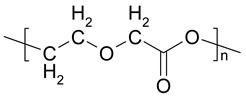 Poly-p-dioxanone	PDS	70%—14 days 50%—28 days 25%—42 days	up to 90 days	180–210 days
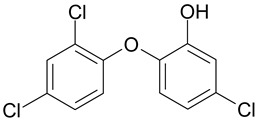 TRICLOSAN Polymer Plus—Antibacterial coating made of triclosan	Monocryl Plus	60%—7 days 30%—14 days	21–28 days	90–120 days
	Vicryl Plus	75%—14 days 50%—21 days 25%—28 days	28–35 days	56–70 days
	PDS Plus	80%—14 days 70%—28 days 60%—42 days	up to 90 days	180–210 days

**Table 2 biomedicines-10-01053-t002:** Changes in the tensile strength with respect to the tensile strength for the initial surgical sutures.

Type of Surgical Suture	Duration of the Experiment (Days)	△R_m_/R_mRef_ (%)				
		Sterile Environment			Contaminated Environment	
		Saline	Pancreatic Juice	Bile	Pancreatic Juice	Bile
**Monocryl**	7	−43.90	−50.61	−47.74	−50.78	−53.62
	14	−60.68	−76.41	−70.06	−88.84	−72.05
	21	−85.10	−96.10	−93.00	−97.90	−94.46
	28	−97.10	--	--	--	--
**Monocryl Plus**	7	−52.70	−50.95	−51.47	−65.02	−57.70
	14	−68.70	−89.54	−81.17	−85.50	−75.07
	21	−84.78	−96.98	−93.34	−96.84	−94.68
	28	−97.62	--	--	--	--
**Vicryl**	7	−13.79	−17.95	−0.78	−44.78	−25.63
	14	−29.01	−41.16	−29.45	−81.91	−45.73
	21	−34.85	−68.02	−57.92	−94.74	−85.39
	28	−78.50	--	−93.21	--	--
**Vicryl Plus**	7	−8.66	−24.65	+2.73	−49.79	−34.83
	14	−13.93	−65.45	−12.45	−77.58	−51.77
	21	−45.93	−81.80	−65.95	−95.44	−87.70
	28	−82.46	--	−96.92	--	--
**PDS**	7	−7.76	−14.08	+0.43	−1.62	−1.71
	14	−3.41	−24.57	−0.68	−1.02	−2.56
	21	−15.36	−25.43	−1.28	−26.96	−8.96
	28	−16.13	−57.94	−11.60	−45.74	−12.71
**PDS Plus**	7	−2.84	−5.32	−8.37	−7.98	−16.04
	14	−8.06	−22.54	−9.47	−19.64	−16.04
	21	−11.35	−35.92	−20.19	−26.29	−16.66
	28	−21.13	−49.92	−24.96	−42.49	−24.57

△R_m_—change in tensile strength with respect to the reference value for individual polymers (R_m_ − R_mRef_). R_mRef_—tensile strength for the initial surgical sutures (reference state—see Appendix A).

## Data Availability

Not applicable.

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
