# Peer review of "Which Suture to Choose in Hepato-Pancreatic-Biliary Surgery? Assessment of the Influence of Pancreatic Juice and Bile on the Resistance of Suturing Materials—In Vitro Research"

_biomedicines, 2022, doi:10.3390/biomedicines10051053_

Round 1

Reviewer 1 Report

this is a very interesting paper, i have to suggest to add a paragraph inton the discussion about the differences between monofilament or not, and if may be differences between  simple interrupted sutures or continous suture

Author Response

Dear Reviewer,

We would like to thank the reviewers for their insightful and valuable comments. We tried to take into account all of them and to correct the manuscript according to the recommendations. We hope that the corrections will improve the quality of the manuscript. We incorporated all the changes and marked them in the manuscript. Reviewers opinions will significantly improve this work, we have expanded the literature of the manuscript with 21 additional publications. The changed fragments in the text are underlined in yellow. The manuscript has been corrected by a native speaker.

Reviewer 1

„Comments and Suggestions for Authors this is a very interesting paper, i have to suggest to add a paragraph inton the discussion about the differences between monofilament or not, and if may be differences between  simple interrupted sutures or continous suture”

Our answer:

As suggested by Reviewer 1, we added to the discussion a section on the differences between monofilament and non-monofilament sutures and between simple interrupted sutures or continuous suture.

Thank you very much for your opinions, we are sure that it will improve this manuscript.

Sincerely yours,

Marcin Gierek

Reviewer 2 Report

The study is an in vitro study of tensile strength of 6 different suture materials (Monocryl, PDS and  Vicryl with and without antibacterial coating) subjected to biliary and pancreatic suspension for up to 28 days. The suspensions were collected from a biliary and pancreatic drain, respectively, in a patient operated for pancreatic cancer. Both sterile and bacterial contaminated juice were investigated.

The study may be relevant but to my knowledge there are no clinical studies indicating that the use of specific suture material in HPB or GI surgery has some significant impact on the complications rates, but this could of course be a potential risk factor.

Suture strength was examined by a tensile strength test that might be an accepted surrogate measure for the risk of anastomotic leakage, but this is only theoretical as many other factors are involved.

The results showed, that at day 7 the tensile strength was similar for Monocryl and PDS in both biliary and pancreatic juice irrespective of antibacterial coating or not. In addition, the strength was similar to a control with the sutures suspended in isotonic saline. The tensile strength in Vicryl was approximately two-fold higher.  At day 21 the tensile strength of the Monocryl suture had decline to a very low value to approximately 15% compared to day 7, whereas the tensile strength in PDS was maintained or only slightly decrease as it was for Vicryl in the sterile milieu but with a large decrease in the bacterial contaminated juice. Results from the bile medium and pancreatic medium was similar.

The authors conclude, the PDS should be preferred for anastomosis in the gastrointestinal tract. Although the results are interesting the authors can not draw this conclusion form the results. Most complications including anastomotic leakage will occur within the first 7 postoperative days, where there were no obvious differences in tensile strength of the sutures, and there are as previously mentioned no published clinical information supporting this conclusion. The authors should comment on the results in relation to the healing process of an anastomosis and the clinical presentation of anastomotic leakage. The longer degradation seen for PDS could this influence the development of anastomotic stenosis?

It is correct, that there will be both biliary and pancreatic juice in the intestine but in a far lower concentration than in the concentrated bile or pancreatic fluid that was obtained in the present study. Have three been made some measurements of the concentration as e.g. amylase or bilirubin in the samples obtained from the patient?

It is unclear for me hove the results from the electron scanning microscopy add to the results. Please clarify.

The manuscript could improve by a professional linguistic revision.

Author Response

Dear Reviewer,

We would like to thank the reviewers for their insightful and valuable comments. We tried to take into account all of them and to correct the manuscript according to the recommendations. We hope that the corrections will improve the quality of the manuscript. We incorporated all the changes and marked them in the manuscript. Reviewers opinions will significantly improve this work, we have expanded the literature of the manuscript with 21 additional publications. The changed fragments in the text are underlined in yellow. The manuscript has been corrected by a native speaker.

Our explanation for Reviewer 2:

In the introduction, we extended the background and we included new references. We also supplemented the references in the discussion.

Referring to the comments of the reviewer, we confirm that only a few papers have been published about the preferred use of specific suture material in HPB or GI surgery and related complications. In the discussion we included 2 additional works: “Pancreaticojejunostomy after pancreaticoduodenectomy: Suture material and incidence of post-operative pancreatic fistula” and “Polyester sutures for pancreaticojejunostomy protect against postoperative pancreatic fistula: a case – control, risk-adjusted analysis (references No. 1 and 14).

Of course, the reviewer is right, in addition to suture strength, many other factors affect anastomotic leak. We have listed the most important of them in the introduction.

We agree with the reviewer's comment that our conclusion “We recommend the use of PDS sutures in intestinal anastomoses exposed to the activity of bile and pancreatic juice” did not result from work and we removed it. Of course, the reviewer is right because most anastomotic leakages occur within the first 7 days after surgery, when no significant differences in tensile strength of the sutures were observed. In the conclusion section, we have only left the conclusions which proceed from the results.

As recommended, in the discussion, we related the results to the healing process and clinical implications. Regarding the reviewer's question about the longer degradation seen for PDS and its influence on the development of anastomotic stenosis, we reply that we found no reports of this issue with regard to any sutures.

Anastomotic stenosis - this complication is most often a consequence of postoperative complications, pancreatic fibrosis, a thin Wirsung duct connected with the intestine, and acute postoperative inflammation of the left pancreatic stump [32]. Stenosis of the pancreatico-enteric anastomosis following pancreatoduodenectomy (PD), a late post-operative complication appearing after a median of 18 (8—120) months [Vanbrugghe C, Campanile M, Caamaño A, Pol B. Management of delayed stenosis of pancreatico -enteric anastomosis following pancreatoduodenectomy. J Visc Surg. 2019; 156 (1): 30-36.doi: 10.1016 /j.jviscsurg.2018.07.009 [33]  According to Table 1 in our work, the period of maintaining the tissue tension of sutures and Absorption period for PDS sutures is significantly longer than for other analyzed sutures. In order to answer the question regarding stenosis, a clinical trial should be conducted to analyze the dependence of the occurrence of stenosis depending on the sutures used. However, it is very difficult due to the low incidence of this long-term complication and the difficulty of eliminating other factors contributing to the formation of stenosis. We found no reports of this complication following long degradation of sutures used for anastomosis.

In our study, we measured amylase and pH in the tested environments. The results are in the Supplementary Material (Table S1). We haven't tested the bilirubin levels.

We would like to clarigy that, in our study in the part of SEM observations we found clear degradation of the structure that can be observed for monofilament sutures made of Monocryl compared to the others. Monocryl sutures exposed to pancreatic juice demonstrated much greater fractures and surface disintegration than in case of bile. In the case of the sutures made of Vicryl, after 21 days of the experiment, a distinct unraveling in the multifilament sutures was observed Karaman et al. obtained similar results [15].

The article has been corrected by a native speaker.

Thank you very much for your important opinions and suggestions, we are sure that it will improve this manuscript.

Sincerely yours,

Marcin Gierek

Round 2

Reviewer 2 Report

Thank you for the changes ´made in the revised version, which answers most of the raised criticism. Especially happy with the conclusion, which now reflect the results from the studies. Several of the arguments in the Discussion section are more speculative than evidence based, but this must be up to the readers to evaluate.  

There are still some minor language issues.

Line 37: What is a puncture of the anastomosis?

Line 59: Where does the evidence for an impaired healing in the anastomosis by bacterial contamination come from.

Line 253: “in which Here” must be corrected

Line 273: What do you mean by the “highest statistical significant” Either modify or give a p-value.

Line 361: External stenting is not possible?

Line 274: Do not understand the text, please modify

This manuscript is a resubmission of an earlier submission. The following is a list of the peer review reports and author responses from that submission.